# Exploring the Potential of High-Voltage Electric Field Cold Plasma (HVCP) Using a Dielectric Barrier Discharge (DBD) as a Plasma Source on the Quality Parameters of Carrot Juice

**DOI:** 10.3390/antibiotics8040235

**Published:** 2019-11-25

**Authors:** Muhammad Umair, Saqib Jabbar, Mustapha Muhammad Nasiru, Tayyaba Sultana, Ahmed M. Senan, Faisal Nureldin Awad, Zhuang Hong, Jianhao Zhang

**Affiliations:** 1National Center of Meat Quality and Safety Control, Synergetic Innovation Center of Food Safety and Nutrition, College of Food Science and Technology, Nanjing Agricultural University, Nanjing 210095, China; umair_uaf@hotmail.com (M.U.); 2018208036@njau.edu.cn (M.M.N.); Ahmedmsenan@njau.edu.cn (A.M.S.); faisalnor@yahoo.com (F.N.A.); 2Food Science Research Institute (FSRI), National Agricultural Research Centre (NARC), Islamabad 44000, Pakistan; saqibjabbar2000@yahoo.com; 3College of Public Administration, Nanjing Agriculture University Nanjing, Nanjing 210095, China; Tayyaba.njau@yahoo.com; 4Quality & Safety Assessment Research Unit, USDA-ARS, Athens, GA 30605, USA; hong.zhuang@ars.usda.gov

**Keywords:** high-voltage cold plasma, enzyme inactivation, coloring compounds, thermal processing alternative, prebiotic carrot juice

## Abstract

The main aim of the current investigation was to contemplate the impact of high-voltage electric field cold plasma (HVCP) on different quality characteristics (enzymes, microbial activities, coloring pigments, ascorbic acid, polyphenolic compounds, °Brix, acidity, and color index) of carrot juice in correlation with thermal processing. A carrot juice (250 mL) sample sealed in pre-sterilized food-grade bottles, which placed between two dielectric quartz plates for HVCP treatment. The gap between the plates was 30 mm, and a stable and uniform plasma dielectric barrier discharge (DBD) generated for 3 and 4 min at 60, 70, and 80 kV. Air was used as a working gas during the DBD-based plasma treatment. The observed rise in temperature was 2–5 °C during the HVCP treatment. A water bath was used to carry out thermal treatment (100 °C for 5 min). The maximum inactivation of enzymes and microorganisms was achieved with thermal treatment and then with HVCP treatment at 70 kV for 4 min. However, maximum retention of coloring compounds, ascorbic acid, total phenols, flavonoids, and tannins was found following HVCP (70 kV for 4 min) treatment compared to thermal treatment. The °Brix, pH, and acidity remained unchanged irrespective of treatments. These findings suggest that HVCP treatment at 70 kV for 4 min may be a good alternative to thermal treatment, and it may successfully be applied in carrot juice production, resulting in reduced enzymes, lower microbial activity, and improved bioactive compounds. The prospects of overcoming the existing conventional physical and chemical methods for sterilization make it a novel and more economical technique to maintain food’s natural nourishment, composition, appearance, structure, and freshness.

## 1. Introduction

The carrot (*Daucus carota* L.) is an important root vegetable with a pleasant taste, high nutritional value, and several benefits to human health due to its specific anti-anemic, anticancer, antioxidant, sedative, and healing characteristics [1]. It contributes to the antioxidant potential due to the presence of significant amounts of bioactive compounds [2]. Carrots are commonly eaten as fresh vegetables; however, due to their perishable nature, they are processed into a variety of products, including juices. During storage, undesirable browning reactions occur that ultimately cause discoloration of the product due to the condensation of phenolic compounds [3]. Enzymes can be involved in the deterioration of food products that cause undesirable color, flavor, or nutritional changes [4]. These enzymes include peroxidase (POD), pectin-methylesterase (PME), polyphenol-oxidases (PPO), and lipoxygenase (LOX). Microorganisms also cause deterioration of fruit and vegetable juices [5]. At present, the shelf life of vegetable juices improved by the inactivation of the enzymes and microorganisms through heat treatments [6]. However, these thermal treatments often cause adverse changes in the food products that account for color alterations, loss of nutrients, and off-flavor development [7]. On the other hand, carrot juice is sensitive to the risk of microbial spoilage due to its low acidity, and it reported that natural micro-flora could play an important role in the deterioration of juice quality. Although high-temperature treatment can decrease the risk of enzymatic and microbial activities, it also causes the loss of heat-sensitive components [8], particularly vitamin C, which is an essential vitamin for the proper functioning of the body, whereby deficiency can cause complications such as scurvy disease.

On the contrary, high-voltage cold plasma (HVCP) non-thermal treatment is a new kind of non-thermal sterilization technique [9] that has more benefits and merits than traditional solvent extraction techniques, while also being more economical from an industry perspective. Because in the case, the equipment operation is simple; furthermore, the technique is non-thermal; it preserves heat-sensitive and water-soluble nutrients and reduces the solvent amount [10]. The prospects of overcoming the existing conventional physical and chemical methods for sterilization make it a novel and economical technique to maintain food’s natural nourishment, composition, appearance, structure, and freshness [11]. Furthermore, HVCP is quick and highly effective, with a low processing cost and low energy input; it was also found to be more efficient for decontamination [12].

Therefore, high-voltage electric field cold plasma (HVCP) treatment can be beneficial, as it was found to stabilize wheat germ by inactivating lipase and lipoxygenase (LOX) enzymes, thereby extending the shelf life of wheat germs [13]. Previous studies showed that HVCP could also inactivate POD in tomatoes [14]. Therefore, the present work was designed with the aim of exploring the effect of HVCP on enzymes such as PPO, POD, PME, and LOX, as well as the levels of polyphenols, ascorbic acid, and coloring compounds (e.g., carotenoids, β-carotene, and lycopene), among other physicochemical properties, in carrot juice; microbial assays (e.g., total plate count, yeast and mold counts) were also carried out. Changes in the final product after treatment with HVCP should be considered for its practical implementation [15]. Carrot juice color and flavor are important quality indicators that attract customers and fulfill customer demand [16]. Some reports showed that HVCP did not affect color, particularly on lightness L* [17]. HVCP-treated pomegranate juice also exhibited a reduction in a* and b* values [18]. However, reductions in L* and b* values were also observed with a concomitant slight increase in a* value after the treatment of apple juice with an atmospheric pressure plasma jet (APPJ) [17]. Thus, the effect on food quality differs significantly depending on the plasma source employed (e.g., APPJ, direct or indirect mode). The influence of the plasma used to sterilize large-scale liquid foods on the quality of fruit juices needs to be evaluated; therefore, it is also important to study the effect on color changes during HVCP treatment.

Moreover, the findings of this study are expected to be beneficial for the solution of quality and safety issues, to attain maximum consumer satisfaction. It is especially important, as consumers are now more conscious about health and diet, and their demand for fresh food brought about increased interest in the mild processing technologies used.

## 2. Material and Methods 

### 2.1. Chemicals

FolinCiocalteu (FC) reagent was bought from Shanghai RyonBiological Technology Co., Ltd., China. Catechin and ascorbic acid were provided by Shanghai Aladdin Bio-chem Technology Co., Ltd. Methanol, hydrogen peroxide, petroleum ether, hydrochloric acid, and n-hexane were provided by Nanjing Chem. Reagent (Nanjing, China). All the other chemicals used in the experiment were of analytical grade.

### 2.2. Procurement of Raw Material, Blanching and Sample Preparation

Raw material (Carrots: *Daucus carota* cv Heitian-5) was bought from the local Suguo supermarket of Nanjing, China. For thermal treatment, good quality carrots (1 kg) were washed using tap water, peeled and then sliced manually to a thickness of approximately 2 cm, and diameter was 10 mm. The sliced carrots were blanched (thermal treatment) in hot water at 100 °C for 5 min. The approximate ratio between the weight of vegetables and water was kept 1:2. Blanched carrots were cooled to room temperature by dipping in cold water, and then these slices were pressed to make juice by using Hurom slow juicer (Model# HU7WN3L). The fresh carrot juice was selected as the control without any treatment. The fresh carrot juice was subjected to HVCP as a design in the experiment.

### 2.3. High Voltage Electric Field Cold Plasma Treatment (HVCP)

The dielectric barrier discharge (DBD) plasma system was used as a reaction cell. The apparatus was composed of an AC dielectric test set (BK-130, Phenix Technologies, Accident, MD, USA), high voltage wires, two circular aluminum electrodes (150 mm diameter), and dielectric barriers (2 mm polypropylene sheets). A high voltage power transformer (CTP-2000K) connected with the voltage regulator (Nanjing Suman Electronics Co., Ltd., Nanjing, China) was used to generate and provide power supply (0–80 kV) to upper and lower electrodes. 

Approximately, 250 mL of carrot juice sample was taken for the treatment of HVCP. The samples were packed in pre-sterilized food grade bottles and were placed between two dielectric quartz plates with a gap of 30 mm and treated at 60, 70, and 80 kV for 3 and 4 min with three-time repetitions and a 30-second break interval in every treatment. Stable and uniform plasma was generated; the air was used as a working gas during the HVCP treatment. The observed rise in temperature was measured by a digital thermometer (2–5 °C) during the HVCP treatment. The schematic diagram of the HVCP treatment for carrot juice has been illustrated in Figure 1.

### 2.4. Residual Enzyme Activity 

#### 2.4.1. Peroxidase Enzyme Residual Activity

The method of [19] was used to determine POD residual activity in carrot juice. Absorbance was measured at 420 nm using a UV-VIS Spectrophotometer (Shimadzu Scientific Instruments Co., Ltd., Beijing, China). The POD residual activity was measured using Equation (1).
Residual activity (%) = 100 × absorbance of treated sample ÷ absorbance of controlled sample(1)

#### 2.4.2. Polyphenol Oxidase Enzyme Residual Activity 

The PPO residual activity in treated and untreated carrot juice samples was determined by the method of [20]. Changes in absorbance were determined at 410 nm after every 60 s, starting from 0 s to 10 min using a UV-VIS Spectrophotometer. PPO residual activity was determined by using Equation (1). All measurements were determined in triplicate. 

#### 2.4.3. Pectin Methylesterase Enzyme Residual Activity

PME is measured as the amount of enzyme that releases 1 mmol of the carboxyl group in 60 s, as described by [21]. A change in absorbance was determined at 620 nm after every 60 s, starting from 0 s to 10 min using a UV-VIS Spectrophotometer. PME residual activity was determined using Equation (1). All the measurements were determined in triplicate. 

#### 2.4.4. Determination of Lipoxygenase Residual Activity

Kim’s method was used to determine the lipoxygenase activity in carrot juice [22]. Changes in absorbance were determined at 234 nm at room temperature using a UV-VIS Spectrophotometer. The oxidation of linoleic acid was calculated using Equation (1). All the measurements were determined in triplicate. 

### 2.5. Determination of Coloring Compounds

#### 2.5.1. Determination of Total Carotenoids

Total carotenoid was measured in all the treatments by a method previously reported in [23] with minor modification. Briefly, the juice sample (25 mL) was poured into a separation funnel with an 80 mL volume of n-hexane/acetone and shaken well. We collected the organic phase after separation. We added 15 mL volume of n-hexane/acetone and shook well again and collected the organic phase. We repeated this step until the aqueous phase became colorless. The organic phase was dehydrated using anhydrous sodium sulfate, and standard β-carotene solutions of different concentrations were prepared for plotting a standard curve. A UV-VIS spectrophotometer was used to measure the absorbance at 450 nm to measure total carotenoids whereas; commercially available β-carotene was used as an external standard (2–10 µg/mL), and data were tested as µg of β-carotene equivalent per milliliter of sample.

#### 2.5.2. Determination of Lycopene Contents

All the samples were treated, and the control was tested to determine the effect of HVCP on lycopene contents. A method explained by [24] was used with slight modification, for the determination of lycopene contents of carrot juice. A juice sample (0.6 mL) was added to 5 mL of BHT in acetone (0.05:99.95, w/v), 5 mL of ethanol (95:5, v/v), and 10 mL of n-hexane of mixture solution. The mixture was then centrifuged for 15 min at 320 g. After shaking, 3 mL of distilled water was added to it. The vial was then agitated for 5 min and held for 2 min to allow the phase separation at room temperature. A UV-VIS spectrophotometer was used to measure the absorbance at 503 nm of the upper n-hexane layer. A pure n-hexane solution was used for the blanked reading. Equation (2) was used to measure lycopene content in carrot juice.
Lycopene contents = (absorbance at 503 nm × MW × DF × 1000) ÷ (ε × L)(2)
where MW is the stand for the molecular weight of lycopene, that is 536.9 g/mol, DF is the dilution factor, L is the path length in cm, and ε is the (172 000 L/mol cm) molar extinction coefficient as reported [25] for lycopene. Lycopene content was determined as µg/mL of the sample. All the measurements were carried out in triplicate.

#### 2.5.3. Determination of β-carotene and Lutein

Briefly, the juice sample (25 mL) was poured into a separation funnel with an equal volume of acetone three times and filtered by a No.1 Whatman filter paper. The filter cake was re-extracted with methanol, and the extract was vigorously mixed with an equal volume of petroleum ether. The upper petroleum ether layer was dehydrated using anhydrous sodium sulfate, and after filtration, it was concentrated using a rotary evaporator at 30 °C. Then, we added 10 mL acetonitrile–methanol–acetone solution (40:40:20, v/v) and kept it at 18 °C in the dark until further analysis. The samples were filtered through a syringe filter (0.45 μm, 13 mm Cat#AS-021345 N-Agela Technologies). Filtered samples were fractionated using Waters Auto Purification high-performance liquid chromatography (HPLC) (Cat. # 1501382671, X Bridge^TM^ prep C18: 5 μm OBD^TM^, 150 mm × 19 mm, Waters, Ireland). The mobile phase was comprised of a mixture solution of acetonitrile-acetone-methanol (40:20:40, v/v), and the flow rate was adjusted to 0.80 mL/min. Commercially available β-carotene and lutein with different concentrations were used to draw a linear regression calibration curve.

### 2.6. Determination of Chlorogenic Acid

Kahle method with minor modifications was used to determined chlorogenic acid [26]. The identification and quantification of chlorogenic acid were determined through HPLC (Waters 600 system). The juice sample was filtered (0.45 μm, 13 mm Cat. # AS 021345-N Agela Technologies), and samples were fractionated through the Waters Auto Purification HPLC column (X Bridge ^TM^ prep C18: 5 μm OBD^TM^, 150 mm × 19 mm, Waters, Ireland). The mobile phase consisted of a mixture solution of aqueous formic acid (A) with methanol (B) (0.1:99.9, v/v). We then injected 5 mL of filtrate sample into the column at a flow rate of 1 mL/minute. A linear biphasic gradient was used with 15–40% solvent B over 10 min, 40–50% over 15 min, 50–75% over 20 min, 75–90% over 25 min followed by a gradual column re-equilibration from 90–65% over 30 min, 65–40% over 35 min, 40–10% over 40 min. Samples were detected using a UV-VIS spectrophotometer at a wavelength of 320 nm, a commercially available chlorogenic acid was used as an external standard, and different concentrations of its solution (10–80 mg/mL) were used to draw a standard curve. Data were presented as mg of chlorogenic acid equivalent per milliliter of samples. 

### 2.7. Determination of Sugar Content in Carrot Juice

Sugar contents were determined by [27]. The identification and quantification of sugar contents were performed using a semi-preparatory high-performance liquid chromatography (HPLC) system (waters 600). The juice sample was filtered (0.45 μm, 13 mm Cat. # AS 021345-N Agela Technologies), and samples were fractionated through the Waters Auto Purification HPLC column (X Bridge ^TM^ prep C18: 5 μm OBD^TM^, 150 mm × 19 mm, Waters, Ireland). The mobile phase consisted of acetonitrile (75:25, v/v) with a 1 mL flow rate. A Cosmosil packed column of D-sugars (4.6×250 mm) was used. Commercially available sucrose, fructose, and glucose were used as external standards. Different concentrations of each solution standard were prepared. Sucrose, fructose, and glucose were used to draw a linear regression calibration curve. Data were analyzed as g of sucrose, fructose, and glucose equivalent per liter of sample.

### 2.8. Phytochemicals Analysis 

#### 2.8.1. Determination of Mineral Contents 

American Public Health Association (1988) [28] described method was followed to determine mineral contents. 1mL of the carrot juice sample was digested with 7 mL of nitric acid solution (65:35, v/v) in Teflon digestion (TFM) vessel. All the conditions for the detection of minerals are given below in Table 1.

We added 1 mL of H_2_O_2_ to this mixture and heated at 200 °C in a microwave oven for 20 min with a one-minute rest after the first ten minutes for better acid digestion. The digested solution was transferred into a 50 mL volumetric flask and diluted up to the mark with pure water. The samples were tested using an inductively coupled plasma-optical emission spectrometer (OPTIMATM 2100 DV, Perkin Elmer Precisely, USA). Commercially available standards of each mineral were purchased and used as standards. All the measurements were conducted in triplicate. We followed the American Public Health Association (1988) described the method to determine mineral contents. 1mL of the carrot juice sample was digested with 7 mL of nitric acid solution (65:35, v/v) in a Teflon digestion vessel (TFM). All the conditions for the detection of minerals are given below in Table 1. 

#### 2.8.2. Total Phenolic Content

Total phenolic content was determined following the method described by [29]. Elution was monitored with a UV-VIS spectrophotometer at 760 nm, and commercially available gallic acid was used as the standard. Different concentration solutions of standard gallic acid were used to draw a standard curve, and the data were depicted as µg of gallic acid equivalent (GAE) per milliliter of sample.

#### 2.8.3. Total Flavonoid Contents

The total flavonoid content was determined using the method described by [29]. The absorbance was measured at 510 nm by a UV-VIS spectrophotometer. Results were expressed as µg of equivalent catechin per milliliter of carrot juice. All measurements were conducted in triplicate. 

#### 2.8.4. Total Tannin Contents

Tannin contents were determined by the method of [30]. The absorbance of all samples was measured by a UV-VIS spectrophotometer at 720 nm using UV-VIS Spectrophotometer (Shimadzu Scientific Instruments Co., Ltd., Beijing, China). Catechin was used as the standard for analysis, and results were expressed as mg catechin per 100 mL of carrot juice.

#### 2.8.5. Determination of Brix

Brix was measured by a hand refractometer (WYT-J 0–32% Chengdu HaochuanGuangdian company, China) at room temperature (25 ± 2 °C). All the measurements were taken in triplicate, and water was used to wash prism of hand refractometer. 

#### 2.8.6. Determination of pH in Carrot Juice

A digital pH meter was used to measure changes in pH in carrot juice samples. Different concentration buffer solutions (pH = 4.0 and 7.0) were used for the calibration of the pH meter. 15–20 mL of each treatment sample was poured into a beaker, and we measured the changes of pH with a pH meter. All measurements were conducted in triplicate.

#### 2.8.7. Determination of Titratable Acidity 

The titratable acidity of the carrot juice samples was measured by the standard method of the Association of Official Analytical Chemists [31]. Briefly, a 15 mL carrot juice sample was taken in a beaker; the volume was completed to 100mL with distilled water. The mix was stirred while titrating against a standard solution of NaOH (4 g/L) to the endpoint (pH 8.2 ± 0.1). The titratable acidity was calculated as citric acid equivalents by Equation (3).
(3)$$Acidity\;\left(\%\right)=\frac{mL\;base\;titrant\times Normality\;of\;the\;base\times Acid\;factor\times 100}{volume\;of\;the\;juice\;sample\;in\;mL}$$

#### 2.8.8. Determination of Color Changes in Carrot Juice

The color was measured using a colorimeter (CR-400, Konica Minolta, Osaka, Japan, with the CIELAB tristimulus parameters: L*, a*, and b*. The L* measures brightness/whiteness of color, a*describes the level of redness (+a*) or greenness (−a*), and b* used to describe the level of yellowness (+b*) and blueness (−b*). A standard white plate (L* = 97.95, a* = 0.89, and b* = 1.23) was used to calibrate the instrument. There are nine samples, and each piece was recorded three times.

#### 2.8.9. Determination of Ascorbic Acid

The method described in [29] was followed to determine the ascorbic acid content in carrot juice. The identification and quantification of ascorbic acid were made through HPLC (Waters 600 system). Filtered samples were separated through a TSKGEL OSD-100Z C18: (4.6 mm× 150 cm × 5 µm) column (Tosoh, Japan). Methanol (30% v/v) was used as a mobile phase at 1 mL/minute flow rate, and samples detection was conducted with UV-VIS spectrophotometer wavelength at 280 nm. Commercially available ascorbic acid, with different concentrations, was used to draw a linear regression calibration curve.

### 2.9. Microbiological Analysis

Microbiological counts were assessed by [32], with slight changes. Serial dilution was performed to maintain the initial microbial load, and the pour-plating method was used to test total plate count (molten agar) and yeast and mold counts (potato dextrose agar). All samples were analyzed in triplicate, and the results were expressed as colony-forming units per gram based on fresh weight (log CFU g^−1^).

### 2.10. Statistical Analysis

The mean and standard deviation values of the results were derived. A complete randomized design (CRD) with one-way Analysis of variance (ANOVA) was applied to calculate the level of significance (*p* < 0.05 the). Least significant difference (LSD) pair-wise comparison test was used to determine the level of significance between means. All the results were analyzed using the Statistical package for social science (SPSS), version 18.0 (SPSS, Chicago, IL, USA).

## 3. Results and Discussion

### 3.1. Effect of HVCP on Enzyme Inactivation

Results indicated that the highest enzyme inactivation was noticed when employing the thermal processing (TP) treatment and followed by HVCP at 70 kV for 4 min (Table 2). Results indicated that less effect was noticed in HVCP at 60 kV for 3 min. A slight change in the effects was observed at 60 kV for 4 min, whereas a significant effect was noticed at 80 kV for 3 and 4 min. However, the effect of HVCP at 70 kV for 4 min was most prominent as compared to other combinations. It suggested that cold atmospheric plasma has the potential to inhibit the formation of alkaline compounds from the decomposition of protein during the storage period through decreasing microbial growth and retarding endogenous protease activity. 

The absorbance charts of the POD and LOX have been drawn to evaluate the impact of different treatments such as Control (Ck) TP, HVCP 60-3, HVCP 60-4, HVCP 70-3, HVCP 70-4, HVCP 80-3, and HVCP 80-4. PME and PPO were drawn by the means value of absorbance at a different time from 0 to 10 min. As given below, the time point enzyme activities chart has been explained in Figure 2.

Albertos has reported the ability of DBD toward reducing the protein decomposition in herring (*Clupea harengus*), which tended to inhibit the formation of alkaline compounds in the early days of a storage period [33]. This effect might decrease the rate of protein decomposition compared to that of the control. A similar phenomenon was observed by [34], where cold plasma treatment was used to study the storage stability of cabbage, lettuce, and dried figs. Therefore, the improved and efficient effect is mainly due to the synergistic interactive effect of reactive oxygen species (ROS) with acidic blanching, and reactive nitrogen species (RNS), which promotes and enhances the overall efficiency of the HVCP treatment.

### 3.2. Estimation of Coloring Compound 

#### 3.2.1. Determination of Chlorogenic Acid

Data regarding the impact of HVCP and thermal processing are presented in Table 3. Results indicate that there is a significant decrease in chlorogenic acid in the case of thermal processing compared with the control. This loss might be due the heat shock as it is reported earlier that some of the polyphenols are sensitive to quick stress and this stress could be because of any source like heat, sound, pressure, and radiations or loss because of diffusion or a leaching effect, as this phenomenon was explained by another scientist [35]. However, in the case of HVCP, a significant increase in chlorogenic acids was observed compared to the control. This increase could be due to the enhanced extraction capacity and maximum disruption of the cell-structure mechanism, which results in a better-releasing capacity of chlorogenic acid that is present in a bound form, and data also showed that this increase was more significant in HVCP treatment (70 kV for 4 min). Thus, the application of HVCP at 70 kV for 4 min can be considered as a novel technique with more benefits and attributes, as it showed potential for better recovery/improvement of the contents of chlorogenic acid in carrot juice that could have lost during traditional blanching treatment for carrot juice processing.

#### 3.2.2. Determination of Total Carotenoids and β-Carotene

The findings of this study have shown that HVCP has involved in the improvement of total carotenoids in carrot juice as compared to control (Table 3). Red-color retention in juices due to better preservation of carotenoids during the juice processing also has been observed in all HVCP treatments; however, this retention was at the peaked when it is treated at 70 kV for 4 min. This better and efficient preservative effect of pigments such as carotenoid and β-Carotene from HVCP at 70 kV may be due to effective inactivation of enzyme activities, mainly because of the inhibitory effect of reactive species (ROS, RNS) produced during HVCP. Similar findings have been reported previously, showing an increase in coloring pigments when apple and carrot juices that were treated with ultra-sonication [29,36], grapefruit juice treated with a pulse electric field [37], and apricot nectar treated with ultra-high hydrostatic pressure [38]. Therefore, HVCP treatment could serve as a potential processing technique for better quality extraction of carrot juice.

#### 3.2.3. Effect of HVCP on Lycopene and Lutein Contents

Lycopene is another key component in all red color fruits and vegetables and has significant importance to consider during the processing conditions. The effect of different HVCP treatments on lycopene and lutein are presented in Table 3.

We have demonstrated that the thermal-treated sample has better retention of lycopene and lutein contents as compared to fresh untreated juice samples. However, the highest increase was observed in HVCP 70 kV for 4 min. In previous reports, it was proven that lycopene and lutein contents were increased significantly when blanched carrot juice was compared with control carrot juice [29]. This better retention of lycopene and lutein contents in blanched samples was due to the weakening of binding forces between the tissue matrix and cell wall that leads to the breakdown of cell structure and improvement in cis-isomerization [39]. This phenomenon is explained by another scientist when the improvement of lycopene and lutein contents during processing was observed, and it was mainly due to the conversion of the trans-isomer form of nutrients to the more bio-available (cis-isomer) form. Ultimately, it enhanced the availability of the lycopene and lutein in the solution [40]. Similarly, an increased in the lycopene and lutein contents were observed in sonication treatment, and this increase might be because of cavitation force and mechanical breakdown, which results in serious chloroplast membrane and cell wall damage [29,36,37]. Therefore, it significantly indicates that HVCP is a promising technique to improve the quality of processed juice by retaining and making its naturally occurring nutrients more bio-available. 

### 3.3. Determination of Sugars Content in Processed Carrot Juice

The effect of different treatments (a combination of various voltages and processing times) on the sugar contents of the processed carrot juice was evaluated. It is depicted that there was a significant decrease in sugar content (sucrose, fructose, and glucose) in thermally processed samples when compared to HVCP and control groups (Table 4). The observed decrease in thermal treatment might be attributed to a poor and insufficient capacity to disrupt sugars and the cell-matrix. Whereas, a slight increase in all sugar content was observed as compared to the control sample in HVCP treatment. It might be due to better releasing and extraction of nutrients. However, the most notable increase was found in HVCP 70 kV at 4 min treatment Table 4. These results were also supported by previous reports, where the effect of blanching on sugar content has been studied for beetroot, carrots, and turnips [41].

Although a full biochemical mechanism that describes this increase is still unknown, it was reported previously that ionized nitrogen species reacts to breaking bonds between sugars and cell membranes. It could be attributed to the electrically charged species from the cold plasma, which may result in the electroporation of cell-matrix and alter the hydrophobic or hydrophilic nature of the membrane and promote the extraction ability [32]. 

### 3.4. Impact of HVCP on the Mineral Profile of Carrot Juice

The result regarding the effect of high voltage cold plasma on the mineral profile (Na, P, K, and Mg) has been demonstrated in Figure 3. A significant reduction was observed in the mineral profile of the thermal sample, while a significant increase in Na, K, and P, and a decrease in Mg were observed in HVCP samples as compared to the untreated controlled sample. A similar effect on mineral content (Na) had been reported previously when the effect of ultrasound treatment on the quality of egg yolk had been observed. Conversely, the findings of that study regarding K, Mg, and P are opposite to our study [42]. Therefore, further research needs to be conducted to explore the exact mechanism that explains the relationship between high voltage and mineral analysis. 

Therefore, an increase in Na, K, and P in all high voltage treated samples is also a solid reason to consider and verify the effect of high voltage cold plasma for the processing of carrot juice. The highest increase was observed in HVCP 70 kV for 4 min when compared to the control sample. Although, in the case of HVCP 60 kV for 4 min, less effect on the mineral profile was noticed. Moreover, a further increase in voltage from 70 kV has a negative impact on mineral contents, and this reduction was more pronounced in the case of Mg.

### 3.5. Effect of HVCP on Ascorbic Acid of Carrot Juice 

The results of TP and other treatments of HVCP have been listed below (Table 5). Results indicated that ascorbic acid is significantly reduced in the TP treatment as compared to HVCP and control. This decrease is mostly attributed to the fact that ascorbic acid is heat-sensitive and can easily be destroyed during the processing above 70 °C that is in following the previous reports [43]. However, a significant improvement was observed in the samples of HVCP treatment. The increased ascorbic acid was because of non-thermal processing, which could be attributed to retaining the heat-sensitive ascorbic acid that is lost because of thermal processing. Further, this increase might be due to the synergistic interactive effect between reactive nitrogen and oxygen species (RONS) and the enhanced intensity of the treatment leading to the release of the bound form of ascorbic acid in the solution and ultimately the enhanced concentration of ascorbic acid in the carrot juice samples. 

Similar findings have been reported when sonication treatment was applied in comparison with thermal processing for apple and carrot juices [29,36]. This increase in ascorbic acid might be due to improved better extraction capacity, which is due to the production of reactive species, which has better penetration ability in the food matrix and can promote the maximum release of nutrients in solution [29].

### 3.6. Effect of HVCP Treatment on Phenolic Compounds 

The different treatment effects on the extraction and determination of phenolics have been explained in Table 5. An obvious decrease was observed in phenolic compounds (total phenols, total flavonoids, and total tannin) in the samples of TP treatment, while a significant increase was observed in the HVCP treatment sample and the combined treatment. The literature about the effect of the plasma application on phenolic compounds is scarce, and only a few investigations have evaluated the changes in phenolic compounds [44].

Similar findings had reported previously when an increase in phenolic compounds was observed, when carrot [29] Saqib et al., 2014b, and apple [36] Abid et al., 2013 juices were treated with sonication treatment, grapefruit juice [37] Aadil et al., (2015) was treated with a pulse electric field, and apricot nectar was treated with ultra-high hydrostatic pressure [38] Huang et al., 2013. In some fruit, like cashew apples, apart from the flavonoids, the phenolic compounds are bonded to the cell membrane and may require a certain level of energy to become free and available (increasing its overall amount in the sample). Therefore, an increase in the level of energy may cause an increase in polyphenols and flavonoids. As, these compounds required different levels of energy, with flavonoids requiring less energy to be released from their bound forms than all the other polyphenols [45] Alves-Filho et al., 2016. 

Increased concentrations of different polyphenolic compounds such as (+)-catechin, (−)-epicatechin, epigallocatechin gallate, quercetin-3-galactoside, and ellagic acid were previously found in pasteurized strawberries [46]. Increased content of phenolic compounds with increasing temperature was reported previously [47] (Odriozola-Serrano et al., 2008. Another study [48] by Rodríguez et al. (2017) reported a positive effect of the treatment time on phenolic compounds when a higher amount of plasma (ionic and singlet species) was fed to the sample chamber (30 and 50 mL/min), but overexposure to the highest flow rate may result in a decrease of the relative total flavonoid content. A possible explanation for these effects has been attributed to enhanced phenolic extraction when juices are treated with cold plasma, chemically reactive species, charged particles, and UV photons are created. They possess sufficient electrical energy to break the covalent bonds and induce numerous chemical reactions, which might increase the cell membrane breakdown and improve the hydrolysis and depolymerization of ellagitannins, which results in an increase of ellagic acid content during plasma treatment [49] Zoran et al., 2016.

A scientist [50] has studied the effect of direct plasma application (70 kV at 50 Hz) during 15, 30, 45, and 60 s on prebiotic orange juice. In this study, the author reported that after 60 s of treatment, the relative total phenolic compounds (TP) of orange juice was found to be 94% and 76% for the juice placed directly below and beside the plasma generator. Similar findings were reported when the gas-phase plasma impact on phenolic compounds in pomegranate juice was studied [49].

Whereas, another researcher [51] Brandenburg et al. (2007) suggested a theory of governing the direct reaction of plasma species with phenolic compounds such as hydroxyl radicals, peroxyl radicals, atomic oxygen, and singlet oxygen. However, the indirect reaction of UV and shock waves might be responsible for increasing the activity of phenylalanine ammonialyase enzymes, which would, in turn, increase the phenolic compounds [52] Stevens et al., 1996. The increase in the phenolic and flavonoid content of blueberries have also been attributed to depolymerization and dissolution of cell wall polysaccharides, which facilitates higher extraction of the conjugated phenolic compounds [53] Sarangapani, et al., 2016b. A possible explanation for these facts has attributed to enhanced phenolic extraction as phenolic compounds in plant materials are mostly linked with plant cell wall polysaccharides, and the interactions of atmospheric plasma reactive species with secondary plant metabolites may disintegrate the phenolic-cell wall matrix bonds [54] Khoddami et al., 2013).

Moreover, the interactions of atmospheric plasma reactive species with secondary plant metabolites in lamb’s lettuce in [55] suggested that OH, O, and O_2_ species may lead to the erosion of the epidermal layer of lettuce where flavonoids and other compounds accumulated in the central vacuoles of guard cells and epidermal cells are released. 

However, overexposure may result in a decrease in phenolic compounds. Once phenolic compounds are in their free form, they are more prone to be degraded by oxidative species on extensive exposure, thus leading to a decrease in quantity [49]. The decrease in phenolics was observed in the previously plasma-treated juices, with no significant influence of plasma parameters. In the case of fresh lettuce leaves, cold plasma treatment decreased the content of all investigated phenolic acids [55]. Moreover, the reduction in TPC and TFC at higher treatment times might be due to reactive species, which were scavenged by phenolic compounds in the blueberries and also the cleavage of the central heterocycle in the polyphenolic skeleton and oligomerization [56].

### 3.7. Effect of HVCP Treatment on pH, °Brix, Acidity, and Color Index 

The different treatment effects on the determination of pH, °Brix titratable acidity, and changes in colors (L*: lightness; a*: redness; b*: yellowness) of total carotenoids have been explained inTable 6.

No significant effect was observed in pH, acidity, and °Brix after treatment with TP and HVCP. This study is under the previous report [57]. The results indicated that TP treatment has significantly improved the color values (L*, a*, and b*) of carrot juice. Furthermore, an increase in L* and a* value has been reported in apricot nectar, when treated at higher pressure as compared to the control sample [38] Huang et al., 2013. A significant decrease was observed in HVCP 60 kV, for 4 min, additionally, L*, a* values were decreased significantly, whereas the b* value was increased in the case of HVCP treatments 70 kV for 3 min. However, HVCP treatments 70 kV for 4 min have shown an obvious improvement in all color index (L*, a*, and b*). Therefore, thermal processing might have tempted in color enhancement. Changes in color might be due to improper activation of enzymes, which results in the enzymatic browning and causes the reduction in a color index. HVCP-treated pomegranate juice also exhibited a reduction in a* and b* values [18]. However, observed reductions in L* and b* values with a concomitant slight increase in a* value after treatment of apple juice with an atmospheric pressure plasma jet, also called APPJ [17].

### 3.8. Antimicrobial Activity Assay

As a key important factor for any processing technology, no technology can be authenticated without significant success in reducing and controlling microbes during the processing. Therefore, this study also elucidated the effect of HVCP on microbial activities (Figure 4). Total plate count (TPC), yeast, and mold counts have been tested in carrot juice. A significant reduction was observed in all HVCP treated samples as compared to the control sample, particularly the HVCP treatment at 70 kV for 4 min, which showed the highest reduction in TPC compared to other treatments. Similarly, significant reducing effects of all the blanched and HVCP treatments were observed while the HVCP treatment for 70 kV 4 min were prominent. Similar findings have been observed by [29,36,37] who reported similar trends for microbial inhibition in their studies when they tested the combined effect of ultra-sonication and blanching for carrot, apple, and grapefruit juices.

This better reduction of microbial activities could be because of the biocidal effect of high voltage that inhibits the growth of microorganisms and enhance the stability and shelf life of the juices. It could be attributed to the production of reactive nitrogen and oxygen species (RONS) during the exposure of plasma generated atmosphere, as shown in Figure 5. It is also found to be following the other previous reports in which the effect of gas-phase surface discharged plasma with a spray reactor for the Zygosaccharomycesrouxii inactivation in apple juice has been studied [58].

## 4. Conclusions

The findings of the recent work showed that HVCP processing has the potential for improving the quality and safety of carrot juice, and it may successfully be employed in beverage industries. Furthermore, the antimicrobial behavior of HVCP treatment has indicated that due to the presence of the reactive species and lethal interaction of hydrogen peroxide and acidified nitrites (NO^−^ and NO_2_^−^), the treatment reduces microbial load and enhances the stability of carrot juice with better nutritional value. Due to this beneficial aspect, it is appropriate for industrial applications for heat-sensitive food items in which heat treatments lose nutrients. Moreover, HVCP treatment had a significant effect on the quality characteristics of carrot juice, such as color, acidity, and odor. Keeping all the above findings and also the certain benefits, i.e., simple and controllable equipment handling, and process operation compared to traditional processing and preservation techniques, we suggest that high voltage cold plasma may successfully be employed in the food industry.

## Figures and Tables

**Figure 1 antibiotics-08-00235-f001:**
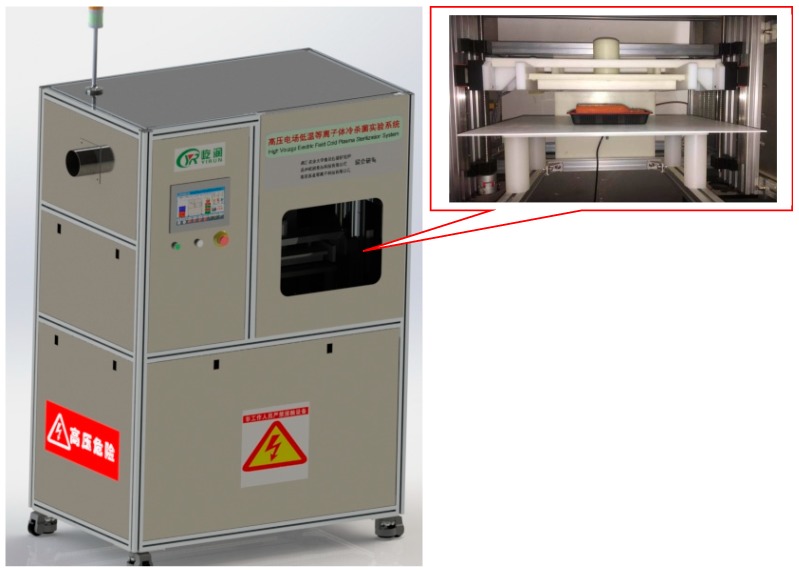
The schematic diagram of the high-voltage electric field cold plasma (HVCP) treatment for carrot juice.

**Figure 2 antibiotics-08-00235-f002:**
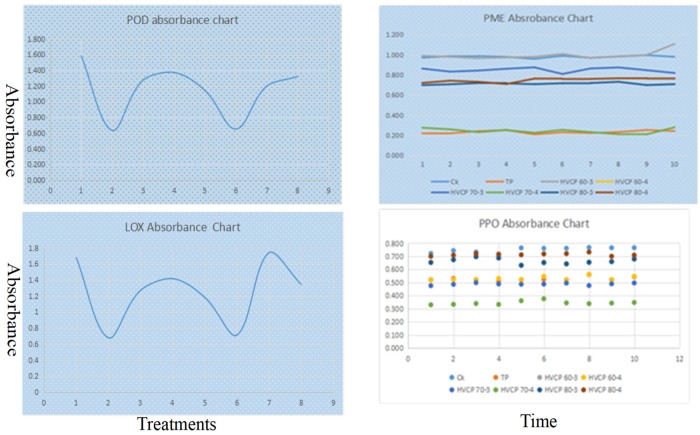
The time-point enzyme activity chart.

**Figure 3 antibiotics-08-00235-f003:**
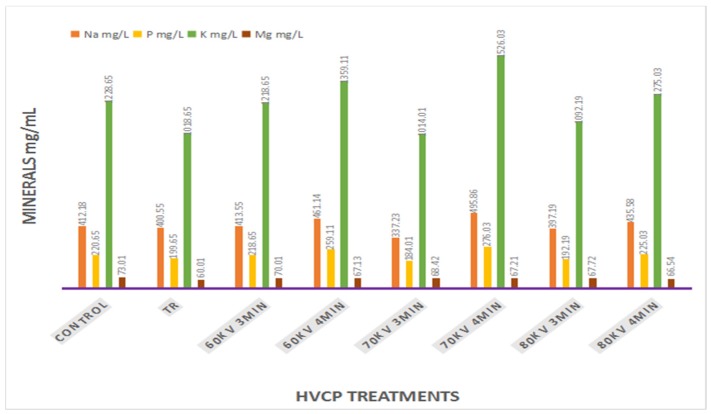
Of HVCP treatment on the mineral profile of carrot juice.

**Figure 4 antibiotics-08-00235-f004:**
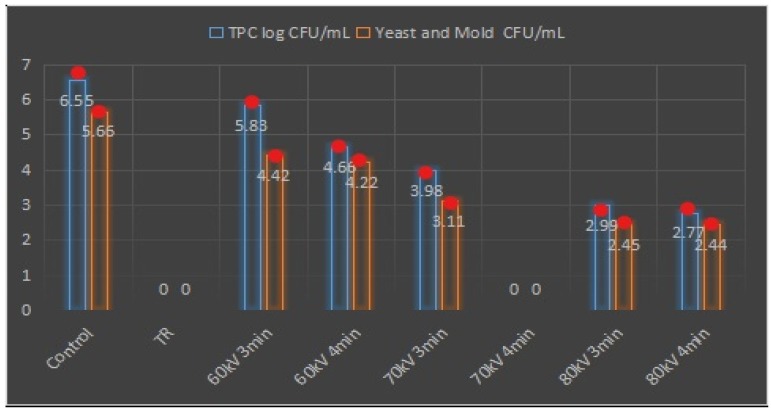
Of HVCP treatment on microbial activity of carrot juice, Noted red dots on the top represents the standard means. Total plate count (TPC) and colony-forming units (CFU).

**Figure 5 antibiotics-08-00235-f005:**
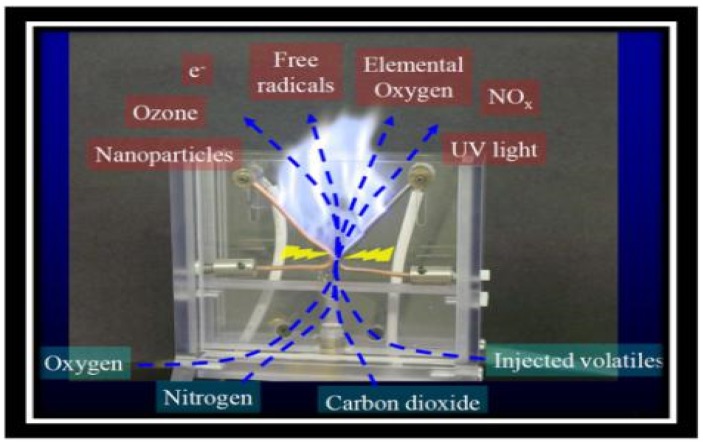
The reactive nitrogen and oxygen species (RONS) during the exposure of plasma.

**Table 1 antibiotics-08-00235-t001:** The working conditions of the plasma-optical emission spectrometer.

Working Conditions	Flow Rate/Units
Elements (Na, P, K, and Mg)	589.5, 213.6, 766.5, and 285.2 nm
Nebulized gas discharge	0.85 L/Min
Plasma gas discharge	16.5 L/Min
Auxiliary gas discharge	0.21 L/Min
Plasma gas discharge	15 L/Min
Sample flow rate	1.8 mL/Min
Operating power	1450 W
View Axial	Interface shear gas
Sample uptake rate	1.25 mL/mint
Spray chamber	cyclonic
Nebuliser type	Meinhard
Nebuliser set up	Instant
Replicates	3 times

**Table 2 antibiotics-08-00235-t002:** The effect of HVCP treatment on enzyme inactivation of carrot juice (*n* = 3).

Treatment	PPO Residual Activity (%)	POD Residual Activity (%)	PME Residual Activity (%)	LOX Residual Activity (%)
Control	100 ± 0.00 ^a^	100 ± 0.00 ^a^	100 ± 0.00 ^a^	100 ± 0.00 ^a^
TP-100-5	11.10 ± 0.01 ^f^	15.23 ± 0.25 ^g^	10.09 ± 0.11 ^f^	13.12 ± 0.32 ^g^
HVCP 60 kV 3 min	77.90 ± 0.20 ^b^	74.40 ± 0.52 ^b^	79.12 ± 0.20 ^b^	81.47 ± 0.12 ^b^
HVCP 60 kV 4 min	59.63 ± 0.43 ^c^	60.22 ± 0.20 ^c^	77.22 ± 0.31 ^c^	72.19 ± 0.12 ^c^
HVCP 70 kV 3 min	30.09 ± 0.53 ^e^	21.19 ± 0.14 ^f^	27.44 ± 0.80 ^h^	31.73 ± 0.13 ^f^
HVCP 70 kV 4 min	11.20 ± 0.09 ^f^	15.73 ± 0.35 ^g^	10.21 ± 0.19 ^f^	13.42 ± 0.21 ^g^
HVCP 80 kV 3 min	30.45 ± 0.40 ^e^	40.50 ± 0.14 ^e^	39.78 ± 0.41 ^e^	43.16 ± 0.65 ^e^
HVCP 80 kV 4 min	40.32 ± 0.72 ^d^	51.34 ± 0.27 ^d^	51.41 ± 0.55 ^d^	63.12 ± 0.22 ^d^

Values with a different letter (a–h) are significantly different from each other (*p* < 0.05). TP-100-5: Thermal processed at 100 °C for 5 min, HVCP 60 kV 3 min: high voltage cold plasma treatment process at 60 kV for 3 min, HVCP 60 kV 4 min: high voltage cold plasma treatment process at 60 kV for 4 min, HVCP 70 kV 3 min: high voltage cold plasma treatment process at 70 kV for 3 min, HVCP 70 kV 4 min: high voltage cold plasma treatment process at 70 kV for 4 min, HVCP 80 kV 3 min: high voltage cold plasma treatment process at 80 kV for 3 min, HVCP 80 kV 4 min: high voltage cold plasma treatment process at 80 kV for 4 min. Note; Polyphenol-oxidases (PPO), peroxidase (POD), pectin-methylesterase (PME), and lipoxygenase (LOX), thermal processing (TP).

**Table 3 antibiotics-08-00235-t003:** The effect of HVCP treatment on the quality parameters of carrot juice (*n* = 3).

Treatment	β-Carotene µg/100 mL	Chlorogenic Acid µg/mL	Carotenoids (µg/mL)	Lycopene (µg/mL)	Lutein (µg/mL)
Control	24.11 ± 0.10 ^d^	22.30 ± 09 ^e^	8.22 ± 02 ^e^	0.52 ± 01 ^f^	1.22 ± 06 ^g^
TP-100-5	20.63 ± 0.25 ^e^	18.67 ± 08 ^g^	7.81 ± 03 ^f^	0.61 ± 09 ^e^	1.35 ± 05 ^f^
HVCP 60 kV 3 min	24.12 ± 0.90 ^d^	21.93 ± 01 ^f^	9.01 ± 09 ^d^	0.91 ± 03 ^d^	1.43 ± 07 ^e^
HVCP 60 kV 4 min	24.21 ± 0.27 ^d^	23.16 ± 02 ^d^	10.03 ± 08 ^c^	1.83 ± 05 ^b^	1.56 ± 09 ^d^
HVCP 70 kV 3 min	25.23 ± 0.69 ^c^	24.04 ± 05 ^c^	9.19 ± 07 ^d^	1.08 ± 06 ^c^	1.51 ± 01 ^d^
HVCP 70 kV 4 min	26.54 ± 0.11 ^a^	27.31 ± 06 ^a^	12.06 ± 05 ^a^	1.93 ± 04 ^a^	2.03 ± 23 ^a^
HVCP 80 kV 3 min	25.19 ± 0.21 ^c^	24.31 ± 06 ^c^	11.23 ± 05 ^b^	1.80 ± 04 ^b^	1.63 ± 04 ^c^
HVCP 80 kV 4 min	25.84 ± 0.12 ^b^	25.31 ± 06 ^b^	11.03 ± 05 ^b^	1.81 ± 04 ^b^	1.76 ± 12 ^b^

Values with a different letter (a–g) are significantly different from each other (*p* < 0.05). TP-100-5: Thermal processed at 100°C for 5 min, HVCP 60 kV 3 min: high voltage cold plasma treatment process at 60 kV for 3 min, HVCP 60 kV 4 min: high voltage cold plasma treatment process at 60 kV for 4 min, HVCP 70 kV 3 min: high voltage cold plasma treatment process at 70 kV for 3 min, HVCP 70 kV 4 min: high voltage cold plasma treatment process at 70 kV for 4 min, HVCP 80 kV 3 min: high voltage cold plasma treatment process at 80 kV for 3 min, HVCP 80 kV 4 min: high voltage cold plasma treatment process at 80 kV for 4 min.

**Table 4 antibiotics-08-00235-t004:** The effect of HVCP treatment on the sugar content in carrot juice.

Treatment	Sucrose g/L	Fructose g/L	Glucose g/L
Control	41.04 ± 01 ^e^	18.65 ± 01 ^b^	20.52 ± 06 ^c^
TP-100-5	36.13 ± 03 ^f^	14.11 ± 02 ^e^	16.11 ± 02 ^e^
HVCP 60 kV 3 min	41.32 ± 02 ^c^	18.01 ± 04 ^b^	20.91 ± 03 ^b^
HVCP 60 kV 4 min	41.88 ± 06 ^d^	18.03 ± 03 ^b^	20.83 ± 04 ^b^
HVCP 70 kV 3 min	42.36 ± 06 ^b^	18.11 ± 02 ^b^	21.99 ± 06 ^a^
HVCP 70 kV 4 min	43.16 ± 03 ^a^	19.31 ± 01 ^a^	21.78 ± 02 ^a^
HVCP 80 kV 3 min	42.92 ± 04 ^b^	16.19 ± 07 ^c^	17.08 ± 04 ^d^
HVCP 80 kV 4 min	41.33 ± 01 ^c^	15.22 ± 02 ^d^	16.23 ± 02 ^e^

Values with a different letter (a–h) are significantly different from each other (*p* < 0.05). TP-100-5: Thermal processed at 100 °C for 5 min, HVCP 60 kV 3 min: high voltage cold plasma treatment process at 60 kV for 3 min, HVCP 60 kV 4 min: high voltage cold plasma treatment process at 60 kV for 4 min, HVCP 70 kV 3 min: high voltage cold plasma treatment process at 70 kV for 3 min, HVCP 70 kV 4 min: high voltage cold plasma treatment process at 70 kV for 4 min, HVCP 80 kV 3 min: high voltage cold plasma treatment process at 80 kV for 3 min, HVCP 80 kV 4 min: high voltage cold plasma treatment process at 80 kV for 4 min.

**Table 5 antibiotics-08-00235-t005:** The effect of HVCP treatment on phenolic compounds.

Treatment	Ascorbic Acid mg/100mL	Total Phenols GAE (µg/g)	Total Flavonoid CE (µg/g)	Tannin CE mg/100mL
Control	24.11 ± 0.10 ^e^	9.77 ± 0.20 ^e^	0.65 ± 0.88 ^f^	14.65 ± 0.13 ^g^
TP-100-5	22.63 ± 0.25 ^f^	8.44 ± 0.11 ^f^	0.66 ± 0.34 ^e^	11.09 ± 0.11 ^h^
HVCP 60 kV 3 min	25.12 ± 0.90 ^c^	10.03 ± 0.9 ^c^	0.86 ± 0.51 ^c^	16.81 ± 0.09 ^d^
HVCP 60 kV 4 min	24.21 ± 0.27 ^d^	9.80 ± 0.17 ^d^	0.65 ± 0.09 ^f^	15.65 ± 020 ^f^
HVCP 70 kV 3 min	24.23 ± 0.69 ^d^	9.83 ± 0.12 ^d^	0.65 ± 0.23 ^f^	15.73 ± 0.86 ^e^
HVCP 70 kV 4 min	24.54 ± 0.08 ^d^	9.87 ± 0.04 ^d^	1.26 ± 0.55 ^a^	18.96 ± 0.50 ^a^
HVCP 80 kV 3 min	25.24 ± 0.67 ^b^	10.32 ± 0.2 ^b^	0.83 ± 0.43 ^d^	17.03 ± 0.99 ^c^
HVCP 80 kV 4 min	25.50 ± 0.84 ^a^	10.45 ± 0.13 ^a^	1.01 ± 0.22 ^b^	17.81 ± 0.21 ^b^

Values with different letters (a–h) are significantly different from each other (p < 0.05). TP-100-5: Thermal processed at 100 °C for 5 min, HVCP 60 kV 3 min: high voltage cold plasma treatment process at 60 kV for 3 min, HVCP 60 kV 4 min: high voltage cold plasma treatment process at 60 kV for 4 min, HVCP 70 kV 3 min: high voltage cold plasma treatment process at 70 kV for 3 min, HVCP 70 kV 4 min: high voltage cold plasma treatment process at 70 kV for 4 min, HVCP 80 kV 3 min: high voltage cold plasma treatment process at 80 kV for 3 min, HVCP 80 kV 4 min: high voltage cold plasma treatment process at 80 kV for 4 min. Where GAE stands for Gallic acid equivalent and CE is for Catechin equivalent.

**Table 6 antibiotics-08-00235-t006:** Effect of HVCP on pH, acidity, °Brix, and color indexes.

Treatment	Brix	Acidity	pH	Color Index		
				L*	a*	b*
Control	7.77 ± 0.20 ^a^	0.10 ± 0.01 ^a^	6.08 ± 0.01 ^a^	35.31 ± 0.24 ^c^	19.53 ± 0.22 ^c^	27.06 ± 0.10 ^e^
TP-100-5	7.77 ± 0.20 ^a^	0.11 ± 0.01 ^a^	6.08 ± 0.02 ^a^	38.83 ± 0.22 ^a^	20.83 ± 0.21 ^b^	34.59 ± 0.11 ^a^
HVCP 60 kV 3 min	7.77 ± 0.21 ^a^	0.10 ± 0.01 ^a^	6.08 ± 0.01 ^a^	36.72 ± 0.34 ^b^	18.58 ± 0.23 ^d^	30.95 ± 0.88 ^b^
HVCP 60 kV 4 min	7.77 ± 0.21 ^a^	0.11 ± 0.01 ^a^	6.08 ± 0.02 ^a^	33.76 ± 0.33 ^e^	17.80 ± 0.24 ^e^	26.35 ± 0.22 ^f^
HVCP 70 kV 3 min	7.77 ± 0.20 ^a^	0.11 ± 0.01 ^a^	6.08 ± 0.02 ^a^	34.51 ± 0.32 ^d^	18.21 ± 0.26 ^d^	29.12 ± 0.12 ^c^
HVCP 70 kV 4 min	7.77 ± 0.22 ^a^	0.11 ± 0.01 ^a^	6.08 ± 0.01 ^a^	38.41 ± 0.25 ^a^	21.04 ± 0.19 ^a^	33.96 ± 0.88 ^a^
HVCP 80 kV 3 min	7.77 ± 0.21 ^a^	0.11 ± 0.01 ^a^	6.08 ± 0.01 ^a^	36.50 ± 0.13 ^b^	16.49 ± 0.22 ^f^	29.19 ± 0.09 ^c^
HVCP 80 kV 4 min	7.77 ± 0.21 ^a^	0.11 ± 0.01 ^a^	6.08 ± 0.02 ^a^	35.51 ± 0.32 ^c^	19.21 ± 0.26 ^c^	28.12 ± 0.12 ^d^

Values with different letters (a–h) are significantly different from each other (*p* < 0.05). TP-100-5: Thermal processed at 100 °C for 5 min, HVCP 60 kV 3 min: high voltage cold plasma treatment process at 60 kV for 3 min, HVCP 60 kV 4 min: high voltage cold plasma treatment process at 60 kV for 4 min, HVCP 70 kV 3 min: high voltage cold plasma treatment process at 70 kV for 3 min, HVCP 70 kV 4 min: high voltage cold plasma treatment process at 70 kV for 4 min, HVCP 80 kV 3 min: high voltage cold plasma treatment process at 80 kV for 3 min, HVCP 80 kV 4 min: high voltage cold plasma treatment process at 80 kV for 4 min.

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
