# Peer review of "Exploring the Potential of High-Voltage Electric Field Cold Plasma (HVCP) Using a Dielectric Barrier Discharge (DBD) as a Plasma Source on the Quality Parameters of Carrot Juice"

_antibiotics, 2019, doi:10.3390/antibiotics8040235_

Round 1
Reviewer 1 Report
All manuscript needs substantial modifications. There are several mistakes also in english language. Many sentences are unclear and not well written
Abstract
line 15: please write the abbreviation as reported for other parameters.
lines 17-22: these sentences describe deeeply the experiments as reported in materials and methods. in an abstract the authors should report the aim of their research and the results.
line 27: "for 4 min.", please remove dot
INtroduction
line 47: should read as "..(Fellows,1994).
line 66: should read as "..have found that DBD has no effect on color particularly on lightness L*..".
line 67-70: the authors should write the references.
Material and methods
lines 90-91: should read as "The slice carrots were blanched (thermal treatment) in hot water at 100 °C for 5 minutes. The approximate ratio between the weight of vegetables and water was kept 1:2".
line 145 please remove "following".
line 154: please explain the minor modification.
line 159: should read as "..were tested.."
line 227-228: please rewrite the sentence. it is unclear.
line 229: should read as "..Different concentration solutions of standard gallic acid were used to draw a the standard curve.."
line 294: please rewrite the sentence, it is unclear.
Figure 2: no comments are reported in text for this figure. Please improve it.
Estimation of coloring compound: not in table 2 but in table 3. Please modify in the text.
Effect of HVCP treatment on pH, °Brix, acidity, and Color index: discussion on color is not exhaustive: please improve.
microbial counts are in the figure 3 or figure 4?
Please add the statistical analysis on microbial activity.
Author Response
All manuscript needs substantial modifications. There are several mistakes also in english language. Many sentences are unclear and not well written
The Comment followed and english of the manuscript has been improved by native speaker.
Abstract
line 15: please write the abbreviation as reported for other parameters.
The Comment followed
lines 17-22: these sentences describe deeeply the experiments as reported in materials and methods. in an abstract the authors should report the aim of their research and the results.
The Comment followed
line 27: "for 4 min.", please remove dot
The Comment followed
INtroduction
line 47: should read as "..(Fellows,1994).
The Comment followed
line 66: should read as "..have found that DBD has no effect on color particularly on lightness L*..".
The Comment followed
line 67-70: the authors should write the references.
The Comment followed
Material and methods
lines 90-91: should read as "The slice carrots were blanched (thermal treatment) in hot water at 100 °C for 5 minutes. The approximate ratio between the weight of vegetables and water was kept 1:2".
The Comment followed
line 145 please remove "following".
The Comment followed
line 154: please explain the minor modification.
The Comment followed
line 159: should read as ".were tested.."
The Comment followed
line 227-228: please rewrite the sentence. it is unclear.
The Comment followed
line 229: should read as "..Different concentration solutions of standard gallic acid were used to draw a the standard curve."
The sentence has been revised
line 294: please rewrite the sentence, it is unclear.
The sentence has been revised
Figure 2: no comments are reported in text for this figure. Please improve it.
The sentence has been revised
Estimation of coloring compound: not in table 2 but in table 3. Please modify in the text.
The sentence has been revised.
Effect of HVCP treatment on pH, °Brix, acidity, and Color index: discussion on color is not exhaustive: please improve.
The sentence has been revised
Microbial counts are in the figure 3 or figure 4?
The sentence has been revised
Please add the statistical analysis on microbial activity.
The sentence has been revised and statistical analysis of microbial activity has been mentioned in the form of standard deviation.
Reviewer 2 Report
The manuscript by professor Jianhao and co-workers describes the effect of high voltage electric field cold plasma (HVCP) on enzymatic, microbial, and chemical properties of carrot juice. The effects of the HVCP treatment were compared with those of the “traditional” thermal treatment. A number of parameters, including enzyme activities (polyphenol oxidase, peroxidase, pectin methylestrase, and lipoxygenase), microbial activities (total plate count, yeast and mould count), amount of colouring pigments (total carotenoids, β-carotene, chlorogenic acid, lutein and lycopene), ascorbic acid, polyphenols, °Brix, acidity, and colour indexes, were determined and considered in this comparative study.
The authors reported that, under the optimized conditions (70 kV for 4 min), the HVCP treatment causes the maximum inactivation of microorganisms and enzymes, together with the maximum retention of colouring compounds, ascorbic acid, total phenols, flavonoids, and tannins. These results are often comparable, or slightly improved, with respect to the thermal treatment. For these reasons, owing to the high inactivation of microorganisms and enzymes and the retention of bioactive molecules, the HVCP treatment might be (only) preliminary considered as an interesting alternative to the “traditional” thermal treatment. Indeed, in my opinion, further studies including several aspects on the safety of the HVCP treatment and its long-time effects on labile and bioactive compounds are required before the application of the method in the food industries. Although disclosing interesting results that deserve to be published, I am quite sceptic about the straightforward application of HVCP in the food industry. Indeed, the generation of reactive species, known to be responsible of oxidative damage, as a direct consequence of the treatment (see comments below) represents a limit of the proposed method. However, as this topic does not fall exactly within my research field, I wonder whether, at the state of the art, are known applications of HVCP in the food industries.
In my opinion, the paper reports interesting results; the study is well designed and the experiments seems to be properly carried out. However, considering the scope of the journal Antibiotics, I believe that the manuscript would reach a more appropriate audience in a food science journal such as Foods. Indeed, dealing with food science and technology and food engineering and production (see scope of booth Foods and Antibiotics), in my opinion, this work fulfils the requirements to be published in Foods after revisions, as suggested below.
The quality of the manuscript can be improved by addressing the following point:
The authors claim that ROS and RNS species formed during the HVCP treatment are responsible of the antimicrobial activity and are involved in the enzymes deactivation. Taking into account the fact that both ROS and RNS represent dangerous species for human health, as, for example, they are known to be related to many inflammatory and degenerative diseases, this reviewer wonders whether the authors evaluated the ROS and RNS concentration in the final product. Furthermore, the above-mentioned oxidative species might react with labile molecules (i.e. carotenoids, L-ascorbic acid, lycopene, polyphenols) leading to their oxidation or decomposition over the time. I would strongly suggest to consider this aspect by monitoring the concentration of the parameters reported in Table 3 after longer times (i.e. 15 days, 1 month, 6 months, or expected shelf-life for this product). The authors should, at least, add a comment on this regard. Page 11 line 377-384: The authors claim that the HPVC treatment is related to an increased sugar contents. I am not convinced that this point represents an advantage from the nutritional point of view. Indeed an higher amount of readily available “free sugars” is not often related with advantages on the customer’s health (i.e. higher and faster insulin releasing and well known consequences, see OMS guidelines). The authors should add a comment on this regard. I would suggest to better comment data reported in Table 5. Regarding the comparison between control sample, thermal treatment and HVCP treatment, I do not see the claimed significant improvement of the amount of the considered parameters as a consequence of HVCP treatment. The are several grammar and typing mistakes that, in some cases make difficult the reading and the comprehension of the manuscript. The whole text needs to be carefully proofread, possibly with the help of a native English speaker . Page 9 line 313: “Table 2” should be replaced with “Table 3” Page 14: Figure 4, please add a caption.Author Response
The manuscript by professor Jianhao and co-workers describes the effect of high voltage electric field cold plasma (HVCP) on enzymatic, microbial, and chemical properties of carrot juice. The effects of the HVCP treatment were compared with those of the “traditional” thermal treatment. A number of parameters, including enzyme activities (polyphenol oxidase, peroxidase, pectin methylestrase, and lipoxygenase), microbial activities (total plate count, yeast and mould count), amount of colouring pigments (total carotenoids, β-carotene, chlorogenic acid, lutein and lycopene), ascorbic acid, polyphenols, °Brix, acidity, and colour indexes, were determined and considered in this comparative study.
The authors reported that, under the optimized conditions (70kV for 4 min), the HVCP treatment causes the maximum inactivation of microorganisms and enzymes, together with the maximum retention of colouring compounds, ascorbic acid, total phenols, flavonoids, and tannins. These results are often comparable, or slightly improved, with respect to the thermal treatment. For these reasons, owing to the high inactivation of microorganisms and enzymes and the retention of bioactive molecules, the HVCP treatment might be (only) preliminary considered as an interesting alternative to the “traditional” thermal treatment.
The sentence has been improved and elaborated in details.
Indeed, in my opinion, further studies including several aspects on the safety of the HVCP treatment and its long-time effects on labile and bioactive compounds are required before the application of the method in the food industries.
Thank you very much respected reviewer for suggesting me the following comments, we are conducting studies based upon the shelf life and the impact of HVCP treatment on the quality parameters and we will publish our work. For this stage we are mentioning some previous studies that explained the application of HVCP treatment in food processing and preservation. There are also some studies will explain the benefits of HVCP on traditional thermal processing with regards of nutritional aspects as well as improvement of certain micro macro nutrients.
Although disclosing interesting results that deserve to be published, I am quite sceptic about the straightforward application of HVCP in the food industry.
Thanks for your suggestion and advice. Our work is based upon the previously reported work of our research group. We have published some work and there are some other research groups as well, those have published some reports which were conducted to explain and evaluate the impact of HVCP treatment in food. As during the HVCP treatment certain reactive species have been produced. Some studies have reported their safety and application for food processing. There are also some reports that explained the short life of these species. Therefore, the reactive species will be disappeared after certain period. We are also working on this aspect to study the lifetime of these reactive species produced during the HVCP treatment. Therefore, the applications of HVCP treatment in food industry are feasible and applicable.
1. Ekezie, F.G.C., Sun, D.W., Cheng, J.H. (2017). A review on recent advances in cold plasma technology for the food industry: Current applications and future trends. Trends in Food Science and Technology, 69, 46–58. https://doi.org/10.1016/j.tifs. 2017.08.007.
2. Lee H, Jung E.K., Myong-Soo C., Sea C.M. (2015). Cold plasma treatment for the microbiological safety of cabbage, lettuce, and dried figs. Food Microbiol. 51, 74-80. ISSN 0740-0020, https://doi.org/10.1016/j.fm.2015.05.004.
3. Pankaj S.K, Misra N.N, Cullen P.J. (2013). Kinetics of tomato peroxidase inactivation by atmospheric pressure cold plasma based on dielectric barrier discharge. Innov. Food Sci. Emerg. 19, 153-157.
4. Tolouie, H., Mohammad ifar, M. A., Ghomi, H., Yaghoubi, A. S., Hashemi, M. (2018). The impact of atmospheric cold plasma treatment on inactivation of lipase and lipoxygenase of wheat germs. Innov. Food Sci. Emerg. 47, 346-352. https://doi.org/10.1016/j.ifset.2018.03.002
5. Xiang, Q., Liu, X., Li, J., Liu, S., Zhang, H., Bai, Y. (2018). Effects of dielectric barrier discharge plasma on the inactivation of Zygosaccharo mycesrouxii and quality of apple juice. Food Chem. 254, 201–207. https://doi.org/10.1016/j.foodchem.2018.02. 008.
6. Ying, W., Zirong, W., Yahong, Y., Zhenpeng, G., Kangquan, G. Tianli, Y. (2019). Application of gas phase surface discharge plasma with a spray reactor for Zygosaccharo mycesrouxii LB inactivation in apple juice, Innov. Food Sci. Emerg. 52, 450-456.
In my opinion, the paper reports interesting results; the study is well designed and the experiments seem to be properly carried out. However, considering the scope of the journal Antibiotics, I believe that the manuscript would reach a more appropriate audience in a food science journal such as Foods. Indeed, dealing with food science and technology and food engineering and production (see scope of booth Foods and Antibiotics), In my opinion, this work fulfils the requirements to be published in Foods after revisions, as suggested below.
Thanks for your suggestion and comments, we have tried to explain the practical implementation of our work in the food industry. And this work will be helpful for the both academia and industry for exploring the new alternates for the better processing and preservation of food and heat sensitive medicine. Therefore, this work can be considered for the publication in Antibiotics journal.
The quality of the manuscript can be improved by addressing the following point:
The authors claim that ROS and RNS species formed during the HVCP treatment are responsible of the antimicrobial activity and are involved in the enzymes deactivation. Taking into account the fact that both ROS and RNS represent dangerous species for human health, as, for example, they are known to be related to many inflammatory and degenerative diseases, this reviewer wonders whether the authors evaluated the ROS and RNS concentration in the final product
Thanks a lot respected reviewer, yes, we have studied the residual effect of these reactive species in food material after exposure to the HVCP. We would like to share our experience that the concentration of these species produced during the treatment or exposure was significantly much higher than the concentration that we have determined after 3-4 hours of exposure and these spices were completely disappeared after 24 hours of the treatment.
Furthermore, the above-mentioned oxidative species might react with labile molecules (i.e. carotenoids, L-ascorbic acid, lycopene, polyphenols) leading to their oxidation or decomposition over the time. I would strongly suggest to consider this aspect by monitoring the concentration of the parameters reported in Table 3 after longer times (i.e. 15 days, 1 month, 6 months, or expected shelf-life for this product). The authors should, at least, add a comment on this regard.
Thank you very much respected reviewer for suggesting me the following comments, as we have mentioned earlier that this is very important aspect to be studied the impact of HVCP treatment during the storage period of the food material so that the safety can be assured. we are conducting studies based upon the shelf life and the impact of HVCP treatment on the quality parameters and we will publish our work. For this stage we are mentioning some previous studies that explained the application of HVCP treatment in food processing and preservation. There are also some studies will explain the benefits of HVCP on traditional thermal processing with regards of nutritional aspects as well as improvement of certain micro macro nutrients.
Page 11 line 377-384: The authors claim that the HPVC treatment is related to an increased sugar contents. I am not convinced that this point represents an advantage from the nutritional point of view. Indeed an higher amount of readily available “free sugars” is not often related with advantages on the customer’s health (i.e. higher and faster insulin releasing and well known consequences, see OMS guidelines). The authors should add a comment on this regard.
We would like to say thanks to the respected reviewer for his valuable comments; this trend has been reported in previous published work where the impact of ultrasonication (non thermal treatment) on the quality parameters of carrot juice was tested by Saqib et al., 2014. Furthermore this is also in accordance with the reported work of Aadil et al., 2015 that has explained the impact of Pulse electric field treatment on grapefruit juice. Saeed et al. (2016) who has reported the effect of sonication on the pear juice processing. Therefore the improvement in sugar contents during the exposure of plasma will be considered as a beneficial effect of HVCP.
I would suggest to better comment data reported in Table 5. Regarding the comparison between control sample, thermal treatment and HVCP treatment, I do not see the claimed significant improvement of the amount of the considered parameters as a consequence of HVCP treatment.
We would like to say thanks to the respected reviewer for his valuable comments; the explanation of Table 5 has been elaborated in the manuscript.
The are several grammar and typing mistakes that, in some cases make difficult the reading and the comprehension of the manuscript. The whole text needs to be carefully proofread, possibly with the help of a native English speaker. Page 9 line 313: “Table 2” should be replaced with “Table 3” Page 14: Figure 4, please add a caption.
The sentence has been revised
Reviewer 3 Report
The manuscript of Umair Muhammad and colleagues deals the potential application of HVCP in trying to improve the qualities of carrot juice.
First of all, I point out my perplexity about the authors' opportunity to send this manuscript to the journal ‘Antibiotics’. The antimicrobial aspect addressed in this manuscript is only marginal. In my opinion this manuscript would be more suitable for a Food Processing journal.
However, I have two sets of observations to make.
1) the manuscript could be interesting if the purpose was solely to add new process knowledge into the extraction and commercialization of carrot juice. Indeed, this information is contained in the manuscript.
2) the manuscript in some points is vitiated by gross errors of writing in English that sometimes compromise understanding the meaning of some sentences.
The manuscript appears to have been written by a novice student who never wrote an article before and, above all, he did not read it carefully before submitting it. I have counted at least twenty citations in the text that have no counterpart in the References list (eg: Shivare et al, 2009; Sharma et al. 2012; Zhang et al. 2005; Talcott anf Howard 1999, and so on).
Similarly, in the list of references, there are some articles that are not mentioned in the text of the manuscript (eg references 9, 10, 20, 21, 22, etc.).
Line 313 and 329 refer to Table 2, while Table 3 should be the correct choice.
On pages 12 and 15 both figures show the caption "Figure 3".
The discussion about the reasons why polyphenols seem to increase with HVCP compared to control is a bit weak. On the other hand, HVCP is known to generate radical species. Therefore one expects that the radical substances present in the juice may somehow decrease rather than increase. This part of the discussion should be reinforced with some reasoning and some references.
Overall, my judgment is as follows: The manuscript contains a series of interesting results, in particular those reported in tables 2, 3, 4 and 5, while those relating to antimicrobial activity are less so, also because the antimicrobial and antiviral properties of the HVCP methodology has been known for a long time and therefore this work does not add anything new to current knowledge.
Finally, it is necessary that the authors also address the economic aspects of this methodology which cannot be ignored into the view to a possible application of the HVCP to food processing industry of carrot juice production.
Author Response
The manuscript of Umair Muhammad and colleagues deals the potential application of HVCP in trying to improve the qualities of carrot juice.
First of all, I point out my perplexity about the authors' opportunity to send this manuscript to the journal ‘Antibiotics’. The antimicrobial aspect addressed in this manuscript is only marginal. In my opinion this manuscript would be more suitable for a Food Processing journal.
Thanks for your suggestion and comments, we have tried to explain the practical implementation of our work in the food industry. And this work will be helpful for the both academia and industry for exploring the new alternates for the better processing and preservation of food and heat sensitive medicine. Therefore, this work can be considered for the publication in Antibiotics journal.
However, I have two sets of observations to make.
1) the manuscript could be interesting if the purpose was solely to add new process knowledge into the extraction and commercialization of carrot juice. Indeed, this information is contained in the manuscript.
Thanks for your suggestion and comments, we have tried to explain the practical implementation of our work in the food industry and this information has been added in the manuscript.
2) the manuscript in some points is vitiated by gross errors of writing in English that sometimes compromise understanding the meaning of some sentences.
The manuscript appears to have been written by a novice student who never wrote an article before and, above all, he did not read it carefully before submitting it. I have counted at least twenty citations in the text that have no counterpart in the References list (eg: Shivare et al, 2009; Sharma et al. 2012; Zhang et al. 2005; Talcott anf Howard 1999, and so on).
Similarly, in the list of references, there are some articles that are not mentioned in the text of the manuscript (eg references 9, 10, 20, 21, 22, etc.).
Thanks for your suggestion and comments, The sentence has been revised and the correct information has been added in the manuscript.
Line 313 and 329 refer to Table 2, while Table 3 should be the correct choice.
On pages 12 and 15 both figures show the caption "Figure 3".
The sentence has been improved and elaborated in details.
The discussion about the reasons why polyphenols seem to increase with HVCP compared to control is a bit weak. On the other hand, HVCP is known to generate radical species. Therefore one expects that the radical substances present in the juice may somehow decrease rather than increase. This part of the discussion should be reinforced with some reasoning and some references.
The sentence has been improved and elaborated in details.
Overall, my judgment is as follows: The manuscript contains a series of interesting results, in particular those reported in tables 2, 3, 4 and 5, while those relating to antimicrobial activity are less so, also because the antimicrobial and antiviral properties of the HVCP methodology has been known for a long time and therefore this work does not add anything new to current knowledge.
The sentence has been improved and elaborated in details.
Finally, it is necessary that the authors also address the economic aspects of this methodology which cannot be ignored into the view to a possible application of the HVCP to food processing industry of carrot juice production.
The sentence has been improved and elaborated in details.
Reviewer 4 Report
Manuscript Recommendation: Accept with minor revisions
The manuscript titled “Exploring the potential of high voltage electric field cold plasma (HVCP) on the quality of carrot juice” explores the effect and application of HVCP as a good alternative to thermal treatment in carrot juice production with reduced enzymes and microbial activity and improved bioactive compounds. This is an important field of study as quality and safety issues directly correlate with consumer satisfaction in food processing industry. This manuscript has been written very well. However, I would recommend accepting this manuscript after minor revisions as the overall framework of the manuscript needs some editing.
Minor Comments
Please include a reference after the following statement in the introduction section: “On the other hand, carrot juice is readily vulnerable to the risk of microbial spoilage because of its low 49 acidity”. Figure legends need to be revised to highlight the summary of the findings. Additionally, they need to have information about who the statistical analysis was done. Figure 3 needs to be revised. The Y axis legend should be included in the figure. Please utilize a software program like GraphPad Prism. The results section should be separate from the discussion section.
Author Response
The manuscript titled “Exploring the potential of high voltage electric field cold plasma (HVCP) on the quality of carrot juice” explores the effect and application of HVCP as a good alternative to thermal treatment in carrot juice production with reduced enzymes and microbial activity and improved bioactive compounds. This is an important field of study as quality and safety issues directly correlate with consumer satisfaction in food processing industry. This manuscript has been written very well. However, I would recommend accepting this manuscript after minor revisions as the overall framework of the manuscript needs some editing.
Minor Comments
Please include a reference after the following statement in the introduction section: “On the other hand, carrot juice is readily vulnerable to the risk of microbial spoilage because of its low 49 acidity”.
The sentence has been improved and elaborated in details.
Figure legends need to be revised to highlight the summary of the findings. Additionally, they need to have information about who the statistical analysis was done.
The sentence has been improved and elaborated in details.
Figure 3 needs to be revised. The Y axis legend should be included in the figure. Please utilize a software program like GraphPad Prism.
The sentence has been improved and elaborated in details.
The results section should be separate from the discussion section.
The sentence has been improved and elaborated in details. We have explained in details and we have tried to explain in appropriate way for the convenience of the readers as there are certain journals which emphasis for combine results and discussion.
Round 2
Reviewer 1 Report
the authors imprved a lot their manuscript. For this reason I suggest to accept the paper.
Author Response
With due respect, sir it is stated that the English editing has been done by native speaker and other comments have been followed and highlighted in the main manuscript.

Reviewer 3 Report
Dear authors, I am not completely satisfied with the answers provided by the authors.
For example, one of my comments was:
The discussion about the reasons why polyphenols seem to increase with HVCP compared to control is a bit weak. On the other hand, HVCP is known to generate radical species. Therefore, some exponents that the juice may somehow decrease rather than increase. This is a part of our discussion of some reasoning and some references.
The authors replied:
The sentence has been improved and elaborated in details.
Actually, the text of the manuscript has remained unchanged with regard to my question and no addition has been made to the text.
Similarly, to another of my comments:
Finally, it is necessary that the authors also address the economic aspects of this methodology which cannot be ignored into the view to a possible application of the HVCP to food processing industry of carrot juice production.
The authors gave the same answer:
The sentence has been improved and elaborated in details.
But, actually, I did not find a precise answer to the comment I raised.
Finally, my observations about the fact that all the citations in the text of the manuscript must find correspondence in the list of References and vice versa has not been completely transposed. In fact, the citations (Martinez-Hernandez 2018, page 7, line 281) and (Garcia et al. 2006, page 10, line 368) are not present in the References list. Furthermore, on page 9, line 315 the authors quote "Albertos has reported the ability ...."; who is Albertos?
Finally, reference n. 16 (Garcia-Graelles et al.) does not appear in the text of the manuscript.
Author Response
Report 2
窗体顶端
Open Review
(x) I would not like to sign my review report
( ) I would like to sign my review report
English language and style
( ) Extensive editing of English language and style required
( ) Moderate English changes required
(x) English language and style are fine/minor spell check required
( ) I don't feel qualified to judge about the English language and style
|
|
|
|
Yes |
Can be improved |
Must be improved |
Not applicable |
|
Does the introduction provide sufficient background and include all relevant references? |
( ) |
( ) |
(x) |
( ) |
|
Is the research design appropriate? |
(x) |
( ) |
( ) |
( ) |
|
Are the methods adequately described? |
(x) |
( ) |
( ) |
( ) |
|
Are the results clearly presented? |
(x) |
( ) |
( ) |
( ) |
|
Are the conclusions supported by the results? |
( ) |
(x) |
( ) |
( ) |
Author Replied
With due respect, sir it is stated that the English editing has been done by native speaker and other comments have been followed and highlighted in the main manuscript. Particularly introduction, abstract and conclusion portion have been improved.
Once again thank a lot for your best effort and important suggestion to improve my manuscript.
